# Active-feedback quantum control of an integrated low-frequency mechanical resonator

Jingkun Guo[1], Jin Chang ®[1], Xiong Yao[1,2,3] & Simon Gröblacher ®[1] ✉

Preparing a massive mechanical resonator in a state with quantum limited motional energy provides a promising platform for studying fundamental physics with macroscopic systems and allows to realize a variety of applications, including precise sensing. While several demonstrations of such ground-state cooled systems have been achieved, in particular in sideband-resolved cavity optomechanics, for many systems overcoming the heating from the thermal bath remains a major challenge. In contrast, optomechanical systems in the sideband-unresolved limit are much easier to realize due to the relaxed requirements on their optical properties, and the possibility to use a feedback control schemes to reduce the motional energy. The achievable thermal occupation is ultimately limited by the correlation between the measurement precision and the back-action from the measurement. Here, we demonstrate measurement-based feedback cooling on a fully integrated optomechanical device fabricated using a pick-and-place method, operating in the deep sideband-unresolved limit. With the large optomechanical interaction and a low thermal decoherence rate, we achieve a minimal average phonon occupation of 0.76 when pre-cooled with liquid helium and 3.5 with liquid nitrogen. Significant sideband asymmetry for both bath temperatures verifies the quantum character of the mechanical motion. Our method and device are ideally suited for sensing applications directly operating at the quantum limit, greatly simplifying the operation of an optomechanical system in this regime.

Observing and utilizing the quantum effects of a macroscopic mechanical resonator is of significant interest in physics. It offers great opportunities to understanding fundamental physics, such as quantum mechanics with massive objects[1,2], and in quantum applications with mechanical resonators, including quantum metrology[3] and tasks in quantum communications[4,5]. However, the mechanical motion is typically overwhelmed by excess classical noise due to its contact with its surrounding thermal bath[6,7], making it challenging to use in quantum applications. Preparing and initializing the mechanical resonator close to its motional ground state reduces the classical noise, thus is a key pre-

requisite for observing quantum behavior of the mechanical system. Several seminal demonstrations of reducing the phonon occupancy below 1 have been achieved over the past years, often in the sideband-resolved limit where the cavity linewidth is smaller than the mechanical frequency[8–16]. These achievements have paved the way for the experimental observation of mechanical quantum behavior[17–20]. In contrast, operating the optomechancial system in the sideband-unresolved regime has received increasing attention recently, in particular for sensing applications[2,21–29]. In such a sensor, the force to be measured is coupled to the displacement of the mechanical resonator, which is then

[1]Kavli Institute of Nanoscience, Department of Quantum Nanoscience, Delft University of Technology, 2628CJ Delft, The Netherlands. [2]Faculty of Physics, School of Science, Westlake University, Hangzhou 310030, P. R. China. [3]Department of Physics, Fudan University, Shanghai 200438, P. R. China. ✉e-mail: s.groeblacher@tudelft.nl

read-out by the optical field. A large bandwidth of the optical cavity allows obtaining the displacement information faithfully, without any significant filtering by the cavity. Furthermore, in applications where a low-frequency mechanical resonator is required or unavoidable due to its large size[7,22,26,29–35], working in the sideband-unresolved regime significantly relaxes the stringent requirement on the optical cavity. For most experiments, low-mass, high-frequency mechanical resonators[8,15] or Millikelvin bath temperatures achieved with dilution refrigerators[9,12] are typically used. The low temperature and high mechanical frequency reduces the initial phonon number $n_{\text{th}} = k_B T/(\hbar\Omega_M)$[21], where $k_B$ is the Boltzmann constant, $\hbar$ is the reduced Planck constant, $T$ is the bath temperature, and $\Omega_M$ is the mechanical resonance (angular) frequency. In contrast, performing cooling on a more macroscopic system where the resonance frequency is lower, or at a higher temperature, is a more demanding task due to the larger initial $n_{\text{th}}$.

Several experimental demonstrations with a variety of trapped or bulk systems have recently reached phonon numbers $\bar{n} < 1$[13,16,36,37] starting from large bath temperatures and with low-frequency resonators. Here, we demonstrate feedback cooling[2,13,16,38–41] of a fully integrated optomechanical resonator with a mechanical resonance frequency of only 1 MHz, an effective mass of 16 pg and an in-plane dimension of 0.5 mm, which is part of a fully integrated optomechanical system and only moderately pre-cooled in a continuous-flow cryostat. Such a system is ideally suited for sensing applications due to its compact size and easy-to-use experimental setup. We measure the displacement of the mechanical resonator using a homodyne measurement, which is processed and sent to a controller that reduces the mechanical motional energy through active feedback control. By operating at sufficiently large feedback gain we observe sideband-asymmetry using an out-of-loop heterodyne measurement. This imbalance in scattering rates between the Stokes and anti-Stokes processes is a hallmark of optomechanical interaction in the quantum regime[41–45]. The asymmetry allows us to independently calibrate the absolute energy measurement of the mechanical mode[16,36,43], which is in good agreement with the inferred phonon number obtained from the calibrated homodyne measurement. This then allows us to perform feedback cooling to reach a minimum phonon number of $0.76 \pm 0.16$ starting from a mechanical mode temperature of 18 K, corresponding to an initial phonon number of $3.6 \times 10^5$. We further

demonstrate cooling from 77 K, where liquid nitrogen is used in the cryostat, reaching a minimum phonon number of $3.45 \pm 0.15$ and sideband-asymmetry is also observed.

## Results

### Integrated high-Q, high-$g_0$ optomechanical device

Our integrated device consists of a "soft-clamped" mechanical resonator inspired by a fractal structure[34] and an optical cavity formed by a photonic crystal. They are assembled using a pick-and-place method. Details of the device structure and the assembly method are discussed in ref. 46. The mechanical resonator is fabricated from a 50-nm-thick high-stress silicon nitride layer. The mechanical structure and the simulated mechanical mode are shown in Fig. 1a. The structure has a high aspect ratio, with dimensions on the 100's μm scale in the in-plane direction. Its fundamental mode oscillates at 1.1 MHz with an effective mass of 16 pg. The mechanical motion couples to the evanescent field of the photonic crystal cavity, which is made from a separate silicon nitride layer, and is placed above the center of the mechanical structure. The designed optical resonance wavelength is ~1550 nm. These two structures are separated by a small gap of 130 nm. We measure a mechanical quality factor $Q_M = 1.8 \times 10^7$ at room temperature, which increases to ~$5.1 \times 10^7$ at 18 K (see Fig. 1b). Due to the small gap, a large optomechanical coupling $g_0$ can be achieved, which is crucial for efficient feedback cooling. As shown in Fig. 1c, d, for the assembled device, we measure the reflectivity and the mechanical frequency shift at different laser frequency. The optical cavity has a linewidth (FWHM, full width at half maximum) of $\kappa/(2\pi) = 8.8$ GHz, including an external coupling of $\kappa_e/(2\pi) = 6.9$ GHz, putting the device firmly in the sideband-unresolved cavity limit $\kappa \gg \Omega_M$. The overcoupled optical cavity is required for a high total detection efficiency $\eta_{\text{det}}$. The change of the measured mechanical frequency at different detuning arises due to the optical spring effect[21], from which we extract $g_0/2\pi = 224 \pm 4$ kHz. With the device, we also achieve a large single-photon cooperativity, $C_0 = 4g_0^2/(\kappa\Gamma_M) \approx 10^3$ at 18 K, manifesting the high single-photon interaction rate compared to the energy damping of the system[40].

As our device is in the sideband-unresolved regime, a measurement-based feedback protocol[39] can be used to reduce the energy in its mechanical motion. The displacement of the mechanical mode is continuously measured and sent to a controller. The

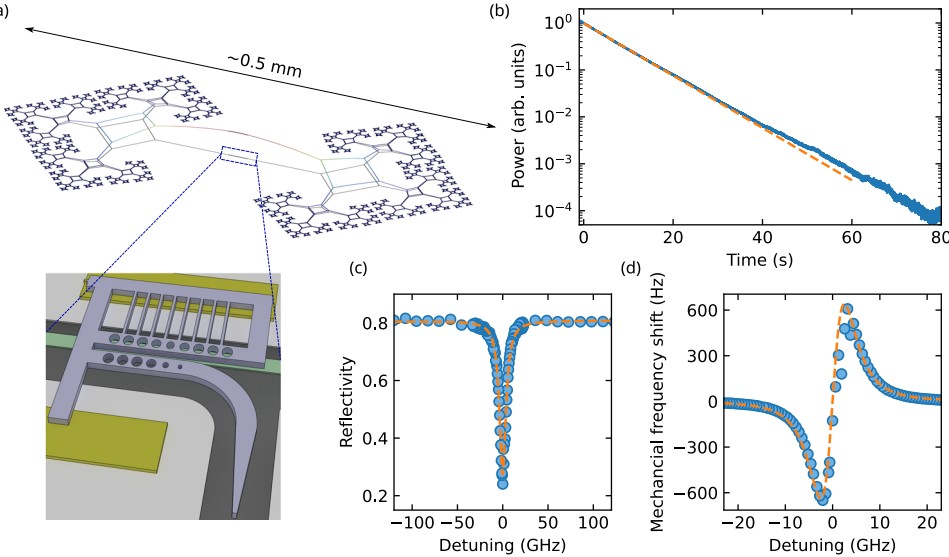

**Fig. 1 | Device design and characterization. a** Mechanical structure and the simulated displacement field of the fundamental mode. At the center (blue box and inset), a photonic crystal cavity (purple) is placed above the mechanics. The spacers (yellow) define the distance between the photonic crystal and the mechanical structure. **b** Ringdown measurement of the fundamental mechanical mode at 18 K. The fit (orange dashed line) allows us to extract a $Q_M = 5.1 \times 10^7$. **c** Cavity reflection and **d** mechanical frequency shift due to the optical spring effect at different laser detuning and at room temperature. A fit to a simple model (orange dashed curves) is used to extract $g_0$.

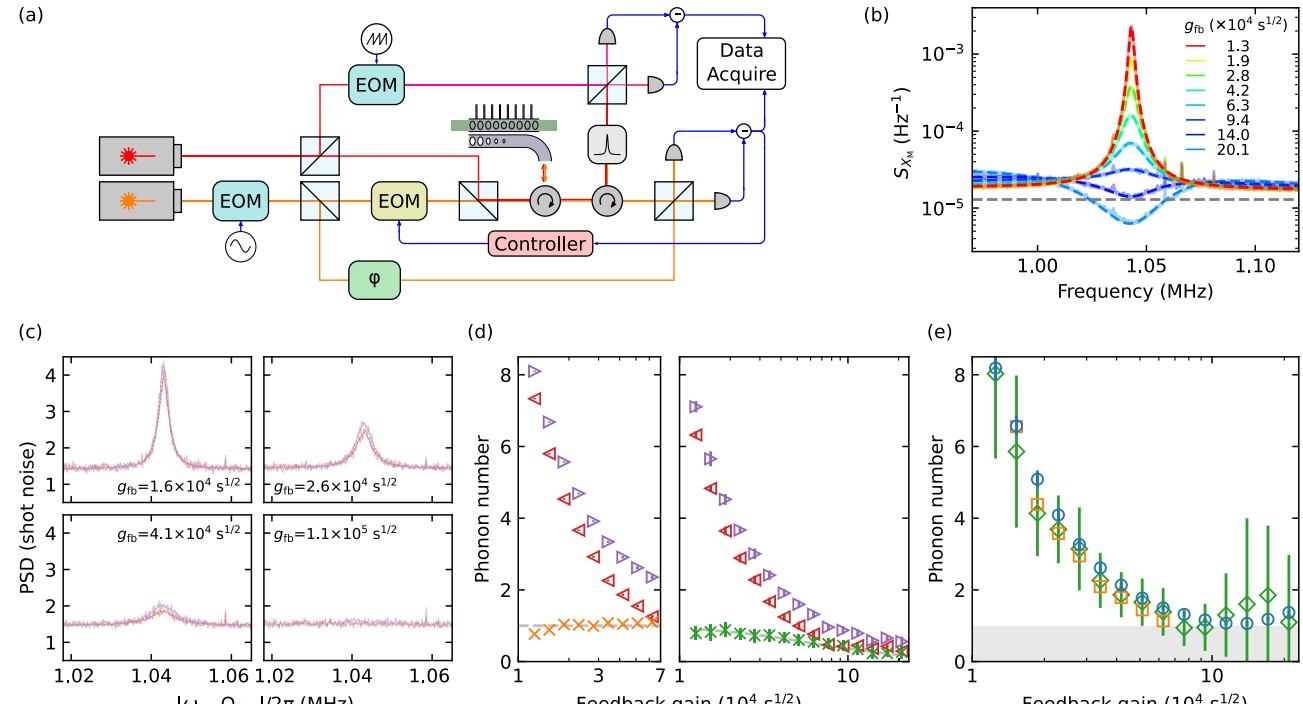

**Fig. 2 | Feedback cooling at 18 K. a** Experimental setup. EOM: electro-optic modulator for phase (blue) and intensity (yellow) modulation. $\varphi$: fiber stretcher for phase locking. In the full setup two sets of measurements are performed simultaneously. The orange lines show the optical paths for the feedback cooling, where the laser is in resonance with the optical cavity, and is modulated by a phase EOM to generate a phase calibration tone. A homodyne scheme is used to measure the phase quadrature of the reflected light. The measured signal is sent to a controller, which controls the light intensity through the intensity EOM. A heterodyne detection scheme (red laser) is used to measure the sideband asymmetry. A phase EOM, driven by a sawtooth signal, is used to shift the optical frequency by $\Omega_{\text{het}}$. **b** Power spectral density (PSD) of the homodyne measurement, converted to the mechanical displacement quanta ($X_M$), at various feedback gains ($g_{\text{fb}}$). The gray dashed line shows the shot noise level, and other dashed curves that fit the data. **c** Power spectral density of the heterodyne signal, normalized to shot-noise, showing two asymmetric sidebands. Integrating the area gives the corresponding phonon number, shown in **d**. The left panel is obtained by fitting the spectrum and integrating the fit, while the data on the right results from directly integrating the measured spectra. Triangles show the sideband power, crosses show the magnitude of the asymmetry, and the dashed line is for the expected asymmetry. The reduction of the asymmetry on the right panel at high feedback gain is due to the finite integration range, which does not fully capture the broadened mechanical spectrum. **e** Phonon number obtained by homodyne measurement (blue), integrating the fit of the sideband-asymmetry spectrum (orange), and the direct integration of the sideband-asymmetry spectrum (green). More details on the data processing can be found in the Methods section. Error bars in all panels represent standard deviations.

controller then processes the information in real-time, and it applies a feedback force onto the mechanical resonator. In optomechanics, this force can be introduced by modulating the optical input power[13,47]. For an optical cavity with large bandwidth, the modulation of the input power is converted to a change of the radiation force[21], without significant filtering effect in the frequency range of interest. In this scheme, realizing a fast and precise measurement is crucial. In a quantum-noise limited measurement with coherent light as the input, a measurement rate is defined to characterize the time-scale at which the zero-point fluctuation can be distinguished from shot noise, $\Gamma_{\text{meas}} = 4\eta_{\text{det}}g^2/\kappa$, where $g = \sqrt{n_c}g_0$ is the multi-photon optomechanical coupling rate with a cavity photon number $n_c$[40,47]. In order to achieve ground state cooling, the measurement rate should be comparable to the decoherence rate, $\Gamma_{\text{dec}} = n_{\text{tot}}\Gamma_M$[40]. Here, $n_{\text{tot}} = n_{\text{th}} + n_{\text{ba}}$ is the total effective bath phonon number, including the excess phonon number due to the back-action noise $n_{\text{ba}} = n_c C_0$[40]. It originates from the quantum fluctuation of the cavity field, which disturbs the mechanical motion through optomechanical coupling. Comparing the two rates,

$$\frac{\Gamma_{\text{dec}}}{\Gamma_{\text{meas}}} = \frac{1}{\eta_{\text{det}}}\left(\frac{n_{\text{th}}}{n_c}\frac{1}{C_0} + 1\right). \tag{1}$$

This ratio should approach 1 for ground-state cooling. The first term compares the thermal decoherence rate to the measurement rate and shows that a high $C_0$ is beneficial, as provided by our device. It reduces the need for a large cavity photon number, which would generate heat

and hence raise the bath temperature due to photon absorption[48–50]. Furthermore, as indicated by the second term due to the back-action noise, there is an ultimate lower bound for the ratio. Any loss in the detection process reduces the precision of the measurement as less information is obtained and therefore a high detection efficiency $\eta_{\text{det}}$ is critical.

## Feedback cooling and sideband asymmetry with LHe

The experimental setup is shown in Fig. 2a. The device is placed in a continuous-flow cryostat, which can be cooled either by liquid helium or liquid nitrogen. A thermometer underneath the sample stage measures the cold-fingers temperature, which reaches 6 K when using liquid helium. The mechanical frequency slightly reduces to 1.045 MHz at low temperature, which corresponds to a thermal phonon occupation of ~$10^5$. A laser, on resonance with the optomechanical cavity, is used for the feedback cooling. A phase EOM, directly after the laser and driven by a sinusoidal signal, generates a phase modulation tone of known amplitude. Both the phase modulation signal and the mechanical motion experience the same transduction due to the optical cavity[51]. This allows us to calibrate the mechanical displacement using the independently obtained $g_0$. A balanced homodyne measurement[52] is set up to measure the phase quadrature of the light, which contains displacement information of the mechanical resonator[21]. The measured signal is then fed to a controller (RedPitaya 125-14), whose output is connected to an intensity EOM and modulating the input light intensity. The controller can realize a

complicated transfer function, which is required for a system with multiple mechanical modes and with significant delay[47]. In this scheme, parasitic detection of the intensity modulation occurs. We have verified that the effect is small, introducing an error of only ~$10^{-6}$ to the feedback force (see Methods section). Another laser, red-detuned from cavity resonance, is used to perform an out-of-loop heterodyne measurement[16,41]. The light is scattered by the mechanical resonator into two sidebands with lower and higher frequency, whose magnitude ratio is given by $\bar{n}/(\bar{n}+1)$, where $\bar{n}$ is the average phonon number of the mechanical mode[9,16,41,45]. The extra quanta in the ratio, which is due to the quantum fluctuation of the mechanical motion[41,43,44], provides an absolute energy scale in our system and can be obtained by measuring the spectrum of the heterodyne signal[16,36]. In our setup, the heterodyne probing light, with both sidebands, is selected by a filter cavity (MicronOptics FFP-TF2, $\kappa/2\pi \approx 35\,\mathrm{MHz}$). It is then combined with the heterodyne local oscillator whose frequency is shifted by $\Omega_{\mathrm{het}}/(2\pi) = 2.81\,\mathrm{MHz}$ with a serrodyne scheme[19,53]. Since our system is in a regime where $\Omega_{\mathrm{M}}/\kappa \sim 10^{-4} \ll 1$, correction of the two sidebands due to the optical cavity[9,16] is not required. Classical amplitude noise might generate spurious sideband asymmetry in the heterodyne detection[41]. In our setup, the heterodyne probe beam merges with the cooling beam only after the intensity EOM, thus the intensity EOM does not introduce additional classical noise and information to the heterodyne probe beam, and the heterodyne detection is unaffected by our cooling modulation. Furthermore, its detuning with respect to the cooling laser ($\gtrsim 1\,\mathrm{GHz}$) is much larger than the detection bandwidth of our photodetector, which eliminates any interference from our feedback control onto the heterodyne probe.

The spectra of the mechanical displacement, obtained from calibrated homodyne measurements, is shown in Fig. 2b. We keep a fixed input power and change the feedback strength by changing the electrical gain of the feedback controller. The feedback introduces an extra damping channel to the mechanical resonator. When the feedback strength is increased the extra damping becomes stronger removing energy from the mechanical motion, manifesting in a broadened mechanical peak with reduced amplitude. At large gain, it enters a noise squashing regime[13] where the measured mechanical signal cancels with the measurement noise. We fit the mechanical spectrum to obtain the average phonon occupancy, as shown in Fig. 2e. The phonon number initially reduces at a small gain, but eventually increases again due to the noise being fed into the mechanical resonator. By fitting the spectra we obtain a minimum phonon occupancy of $1.06 \pm 0.06$. The heterodyne measurement is shown in Fig. 2c, where the power spectral density is normalized to the shot noise level. We clearly see a difference in the amplitude of the two sidebands. We fit the two sidebands to extract the corresponding energy. The result is plotted in the left panel in Fig. 2d, where all the values are normalized by the average energy difference of all curves. The energy difference remains stable over different feedback strengths. At high gain, the mechanical peak is broad and its amplitude is small, which makes fitting no longer possible. We also directly integrate the measured heterodyne spectrum over a frequency range between 1.035 MHz and 1.05 MHz, around the mechanical central frequency. We then subtract the noise floor obtained from the fit. The result is shown in the right panel of 2d. Since the mechanical peak broadens at a larger gain, the energy inside the integration range reduces and thus the energy difference of the two sidebands reduces at larger gain. It nevertheless agrees with the theoretical expectation from the broadening of the peak. We compare the phonon number calibrated from the sideband asymmetry to the phonon number obtained by fitting the homodyne measurement in Fig. 2e, and find both methods to be in good agreement.

### Ground-state cooling with improved detection efficiency
As indicated by Eq. (1), achieving a low $\Gamma_{\mathrm{dec}}/\Gamma_{\mathrm{meas}}$ requires a high detection efficiency $\eta_{\mathrm{det}}$. Performing two sets of measurements

simultaneously requires additional optical components to separate the measurement results, which introduces losses. Furthermore, an out-of-loop heterodyne measurement requires sending additional light into the optical cavity, which leads to larger quantum back-action noise. This is equivalent to a lower $\eta_{\mathrm{det}}$ as the light for the heterodyne measurement does not contribute to the feedback cooling. Still, the quantum fluctuation of the light perturbs the mechanical resonator, leading to an increase of the motional energy. As we have confirmed the agreement for the extracted phonon number between the homodyne measurement and the sideband asymmetry measurement, which shows the validity of our calibration of the in-loop homodyne measurement, we performed a feedback cooling measurement without the sideband asymmetry measurement setup. Extra optical components are removed, as shown in Fig. 3a, to increase the detection efficiency. We fix the cavity photon number to $n_c \sim 350$. With a detection efficiency of $\eta_{\mathrm{det}} \approx 0.49$, we have $\Gamma_{\mathrm{dec}}/\Gamma_{\mathrm{meas}} \approx 2.7$. We measure and fit the spectrum for different feedback strengths (see Fig. 3b), and extract the phonon number, shown in Fig. 3d. In these measurements a minimum phonon number of $0.76 \pm 0.16$ is achieved. We note that the fitted effective bath temperature, with the contribution from the back-action noise $n_{\mathrm{ba}}$ subtracted, corresponds to a thermal temperature of 18.3 K (cf. Fig. 3c), which is significantly higher than the readout from the thermometer. Such a discrepancy is indeed quite common due to the heating of the mechanical structure and non-ideal thermalization of our chip[8]. For comparison, a temperature of 6 K would allow a minimum phonon number of 0.45.

### Feedback cooling with LN$_2$
Cooling a mechanical resonator to close to its motional ground state and revealing quantum effects at a higher and more accessible temperature is technologically of utmost importance but also significantly more challenging. Here we demonstrate feedback cooling starting at 77 K, where the bath is pre-cooled by flowing liquid nitrogen. We otherwise use the same setup as shown in Fig. 2a. The spectrum in the homodyne measurement is shown in Fig. 4a. At higher temperature, the thermal decoherence rate is higher and thus a stronger measurement is needed to achieve a low phonon occupancy. However, still within the regime $\Gamma_{\mathrm{meas}} \ll \Gamma_{\mathrm{dec}}$, the measurement noise floor increases. This is a direct result of the mechanical motion of the low $Q_{\mathrm{M}}$ mode of the photonic crystal[46], which shows a strong $1/f$ feature[6,54] and affects the measurement of the low-frequency mechanical mode. In addition, the initial mechanical quality factor is lower at this elevated temperature with $Q_{\mathrm{M}} = 4.1 \times 10^7$, limiting the minimum achievable phonon number. Nevertheless, sideband-asymmetry (see Fig. 4b), as a unique feature of quantum physics[41–44,55], is still observed at this higher temperature. We extract and compare the phonon occupancy obtained by fitting the calibrated homodyne spectrum, and the heterodyne spectrum and by directly integrating the area of the heterodyne spectrum. The results are presented in Fig. 4c, showing consistency among the methods. The minimal phonon number of $3.45 \pm 0.15$ is determined from the fitting of the homodyne measurement.

### Discussion
We have fabricated an integrated optomechanical device, achieving large optomechanical coupling ($g_0/(2\pi) = 224$ kHz) and a high mechanical quality factor ($Q_{\mathrm{M}} = 5.1 \times 10^7$) for the fundamental out-of-plane mechanical mode at 1 MHz. We pre-cool the bath with liquid helium to an effective mode temperature of ~18 K, where the corresponding thermal phonon occupation is ~$3.6 \times 10^5$, and perform measurement-based feedback cooling to reduce the motional energy. Using sideband asymmetry, we verify our measurement scheme, which agrees well with the result obtained from the calibrated in-loop homodyne measurement, confirming the validity of our calibration. As this double measurement scheme reduces the detection efficiency we obtain an occupation of $1.06 \pm 0.06$, close to the ground state, compared to $0.76 \pm 0.16$, when the setup's efficiency is improved. We

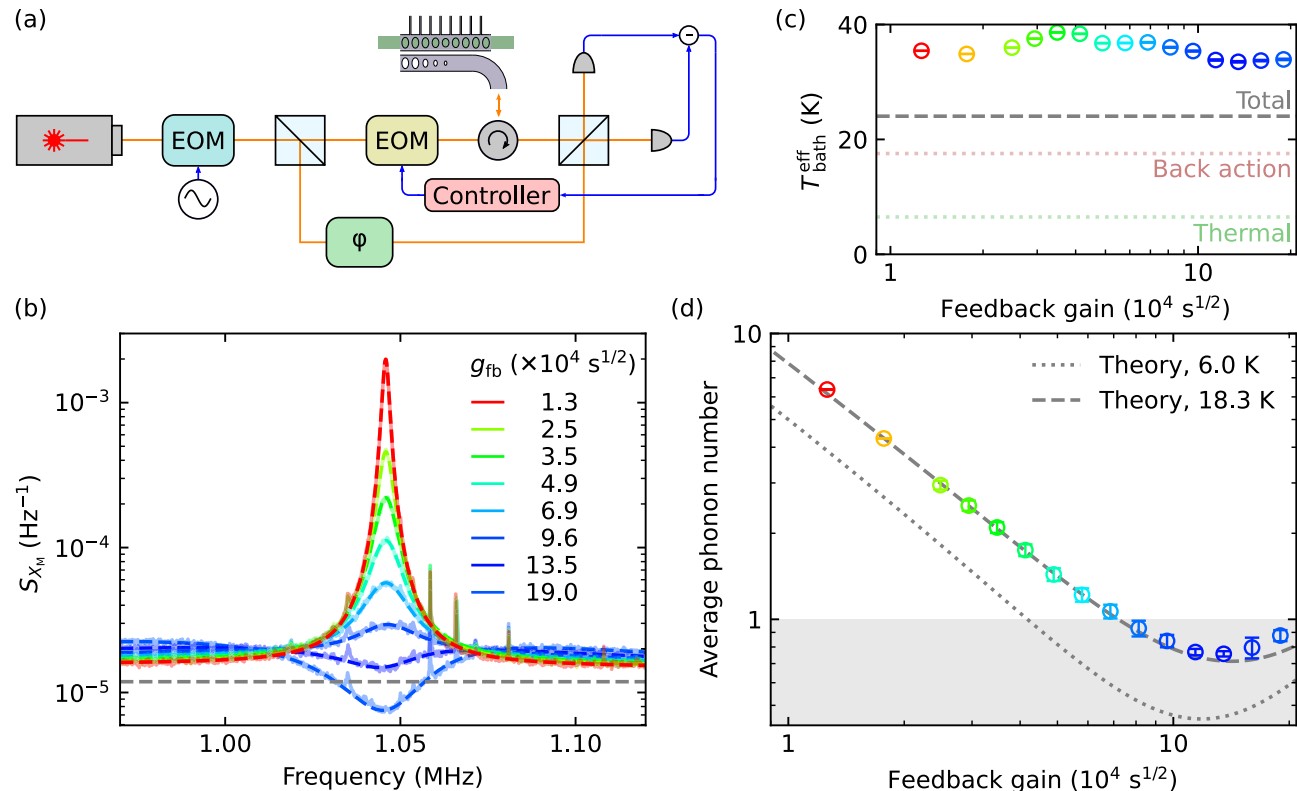

**Fig. 3 | Feedback cooling without sideband-asymmetry measurement.**
**a** Feedback cooling setup. A single, slightly red-detuned laser is used to perform the measurement. **b** A measured spectrum, converted into the displacement quadrature quanta, of the mechanical resonator, at different feedback strength. The fits (dashed lines) are used to extract the system parameters and the phonon occupancy. The extracted effective bath temperature (**c**) is higher than the expected value, indicating excess decoherence (e.g., higher bath temperature). **d** Shows the extracted phonon number, where the gray region shows an occupancy below 1. Error bars in all panels represent standard deviations.

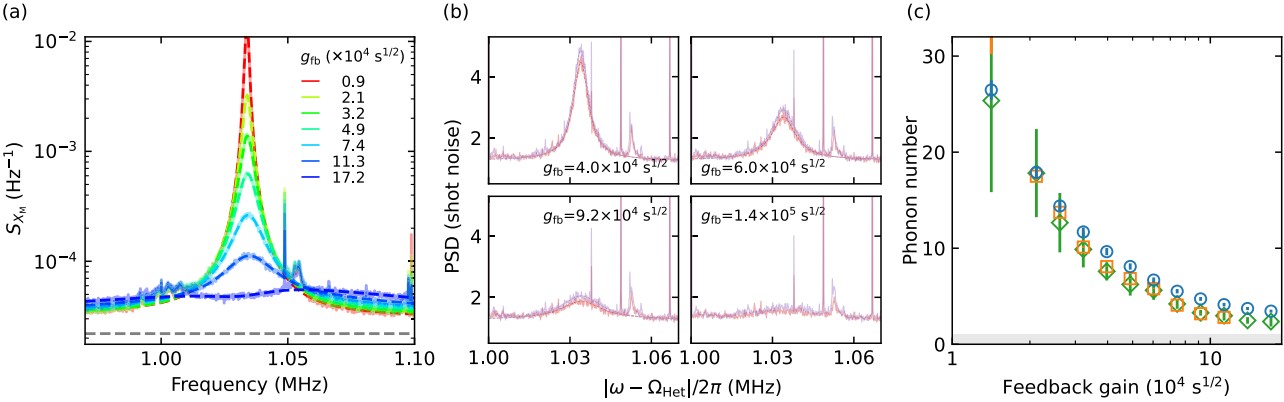

**Fig. 4 | Feedback cooling from an initial bath temperature of 77 K, pre-cooled with liquid nitrogen. a** Power spectral density of the mechanical displacement for various feedback gains. **b** Heterodyne spectrum signal, normalized to the shot noise, for various feedback gains. The asymmetry in the two sidebands at a large gain is clearly visible, where the Stokes scattering (red trace) has a higher power spectral density. Extra peaks are present due to the mixing with higher order modes in the detection[68]. **c** Average phonon number extracted from the homodyne (blue) and from the heterodyne measurement (orange: integrating the fit of the heterodyne spectrum; green: integrating the heterodyne spectrum directly).

further demonstrate that even starting from 77 K, which is only a factor of 4 from room temperature, our technique allows us to reach a regime with significant asymmetry in the two motional sidebands, underlining the quantum nature of the created state. Small improvements in the mechanical quality factor will allow it to directly reach the ground state even from ambient temperatures. The use of our fully integrated optomechanical device, deep in the sideband-unresolved regime, and our simple experimental setup with the ability to cool its fundamental mode into its motional ground state is highly relevant for real-world

applications and will allow for using this technology as an easy-to-use quantum technology. This could include, for example, quantum transducers operating at elevated temperatures, where typically thermal noise contributions overwhelm any quantum signal. Laser cooling the mechanical intermediary would alleviate these limitations[56]. Further applications could include next-generation sensors, such as measuring force and displacement, electro-magnetic waves at radio frequency, temperature, and decoherence[2,7,22,25,26,57–61], as well as for preparing a macroscopic mechanical resonator in a non-classical

state[1,62,63], and other quantum information processing applications[56,64–66].

## Methods

### Experimental setup

We use a Santec TSL-510 laser as our cooling laser, and a New Focus TLB-6728 as the heterodyne probe laser. In both homodyne and heterodyne setups, the path lengths of the local oscillator and the signal arm are matched. We confirm this by scanning the laser over a broad wavelength range and observing the interference fringes. An additional free-space path is added to the homodyne local oscillator to fine-tune the optical path length.

In our experiment, a phase modulation tone is generated to calibrate the mechanical displacement[51]. The phase modulation $\delta\phi_{PM} = \phi_0 \cos\Omega_{PM}t$ is equivalent to a frequency modulation, $\delta\omega_L = \frac{d\phi_{PM}}{dt} = -\phi_0\Omega_{PM}\sin\Omega_{PM}t$. Here, $\phi_0$ is the phase modulation depth, which is much smaller than $2\pi$. Note that the optical cavity transduces both the laser frequency modulation and the mechanical displacement to the phase quadrature. By comparing the homodyne signal, the laser frequency modulation due to the phase EOM is compared to the cavity frequency modulation due to the mechanical resonator, and the mechanical displacement is calibrated. Due to the large optical linewidth of the optomechanical cavity, the resulting transduction to the phase quadrature is very small ~1 MHz. It is then susceptible to the residual amplitude modulation from the phase EOM and the residual amplitude detection in the homodyne measurement scheme. We, therefore, increase the frequency of the phase modulation tone to 60 MHz. While it is still much smaller than the cavity linewidth, the corresponding frequency modulation is larger, and therefore the transduction is larger. It allows us to unambiguously measure the phase modulation. In the detection, the frequency responses of electronics are not flat across the large frequency range. We normalize all the measured values to the optical shot noise, which is measured by blocking the signal arm. Since the shot noise is flat in spectrum, the measured spectra in different frequencies are calibrated and comparable.

### Data processing

In the experiment, we measure the spectrum of the photodetector voltage. In order to account for any filtering effect from the photodetector and the electronics, we normalize all the spectra to the spectrum corresponding to the shot noise. The shot noise spectrum is measured by blocking the light entering the optomechanical cavity and leaving the other experimental parts unchanged.

For the cooling and the homodyne measurement, it is convenient to model our system in a semi-classical way[13,50]. The displacement of the mechanical resonator in the frequency domain and normalized to the zero-point motion is[67]

$$X_M(\omega) = \chi_M(\omega)F(\omega), \tag{2}$$

where the susceptibility

$$\chi_M(\omega) = \frac{\Omega_M}{(\Omega_M^2 - \omega^2) - i\Gamma_M\omega}. \tag{3}$$

$F(\omega) = F_N(\omega) + F_{fb}(\omega)$ is the input "force", consisting of input noise noise and the feedback control. The input noise has multiple origins. The thermal noise, in the high temperature regime ($k_B T/(\hbar\Omega_M) \sim 10^5 \gg 1$ in our experiment), the spectrum is flat[67]

$$S_{F_{th}}(\omega) = 2\Gamma_M(2n_{th} + 1). \tag{4}$$

In addition, the mechanical resonator experiences quantum fluctuation of the optical field (quantum backaction noise, $F_{ba}$)[21].

Our system is in the sideband-unresolved limit, $\kappa/\Omega_M \sim 10^4 \gg 1$, the spectrum is also white[21]. It is therefore not possible to distinguish $F_{ba}$ and $F_{th}$ directly in the experiment. We introduce an effective bath temperature to include the effect of the backaction noise, $T_{eff} = T_{th} + T_{ba}$, and

$$S_{F_N}(\omega) = 2\Gamma_M(2n_{bath} + 1), \tag{5}$$

where $n_{bath} = k_B T_{eff}/(\hbar\Omega_M)$. Since our system is in the sideband-unresolved limit, the optical cavity does not have a filtering effect in the control of the mechanical resonator. The feedback force is directly proportional to the output of the electronic controller with a transfer function $H_{fb}$,

$$F_{fb}(\omega) = g_{fb}H_{fb}(\omega)X_M^{meas}(\omega). \tag{6}$$

Here, $H_{fb}(\omega) = \tilde{H}_{fb}(\omega)e^{i\omega\tau_{fb}}$ includes signal transmission and processing delay $\tau_{fb}$, and $\tilde{H}_{fb}(\omega)$ is the transfer function without time delay. For a sideband-unresolved system,

$$X_M^{meas}(\omega) = X_M(\omega) + X_M^{imp}(\omega) + \epsilon_{fb}gH_{fb}(\omega)X_M^{meas}(\omega). \tag{7}$$

The readout of the actual displacement of the mechanical resonator is not filtered by the optical cavity, giving the first term. The second term is the imprecision of the measurement, including the quantum fluctuation of the measured optical field (vacuum noise)[13,67] and any possible classical measurement noise. We also consider the third term, which is due to the imperfection of our experiment. In our experiment, the feedback controller modulates the intensity of the input light, and we measure the phase quadrature of the light. The modulation and the readout belong to the same optical mode, but they are on different quadratures. However, experimentally, it is challenging to keep the modulation and the measurement perfectly orthogonal. A small error results in measuring a small amount (quantified by a free parameter $\epsilon_{fb}$) of the modulation signal. Experimentally, before performing the cooling, we generate an amplitude modulation tone from the feedback controller, and we fine-tune the homodyne phase locking such that the detected power of the modulation tone is minimized. This guarantees an "almost" perfect detection, where the detected quadrature is orthogonal to the modulation quadrature. In the fit (see below) we obtain $\epsilon_{fb} \sim 10^{-6}$.

Experimentally, the power spectral density (normalized to the shot noise level) is measured. Combining the above equations,

$$S_{X_{meas}}(\omega) = \frac{|\chi_M(\omega)|^2 S_{F_N} + S_{X_M^{imp}}}{|1 - g_{fb}(\chi_M(\omega) + \epsilon_{fb})\tilde{H}_{fb}(\omega)e^{i\omega\tau_{fb}}|^2}. \tag{8}$$

By using $S_{F_N}$, $S_{X_M^{imp}}$, $g_{fb}$, $\epsilon_{fb}$, and $\tau_{fb}$ as fitting parameters to fit the experimental curve, we can extract their values. The spectrum of the actual displacement can be inferred

$$S_X(\omega) = \frac{|\chi_M(\omega)|^2 \left(|1 - \epsilon_{fb}H_{fb}(\omega)|^2 S_{F_N} + |g_{fb}H_{fb}(\omega)|^2 S_{X_M^{imp}}\right)}{|1 - (\epsilon_{fb} + g_{fb}\chi_{fb})H_{fb}|^2}. \tag{9}$$

The effective average phonon number is the integral of the inferred mechanical spectrum[39]

$$\bar{n} = \frac{1}{2}\left(\int_0^\infty \frac{d\omega}{2\pi}\left(1 + \frac{\omega^2}{\Omega_M^2}\right)S_X(\omega)\right) - \frac{1}{2}, \tag{10}$$

which is evaluated numerically.

Our heterodyne measurement is performed by shifting the frequency of the local oscillator beam by $\Omega_{het}$. It is then possible to access

the two sidebands resulted from the optomechanical interaction[16,36,41,42]. For our out-of-loop heterodyne detection, we fit the spectrum of the two sidebands by

$$S(\omega) = \sum_{j=l,r} \left( k_j |\chi_{\text{eff}}(\omega_j)|^2 + n_j \right),$$ (11)

where l, r are for the left and right sidebands. $k_j$ represents the magnitude of the sideband, and $n_j$ is for the noise floor at the vicinity of the sideband. The spectral frequency of the two sidebands are shifted due to the frequency difference on the local oscillator, $\omega_l = \Omega_{\text{Het}} - \omega$, and $\omega_r = \omega - \Omega_{\text{Het}}$ and are fitting parameters. The susceptibility is approximated to have the same form as the susceptibility of a bare mechanical resonator,

$$\chi_{\text{eff}}(\omega_j) = \frac{\tilde{\Omega}_M}{(\tilde{\Omega}_M^2 - \omega_j^2) - i\tilde{\Gamma}_M \omega_j},$$ (12)

where the effective resonance frequency $\tilde{\Omega}_M$ and the effective mechanical dissipation rate $\tilde{\Gamma}_M$ is introduced due to the effect of the feedback cooling, and they are left as fitting parameters. From the fitting, it is then possible to obtain the phonon occupation number

$$\bar{n} = \frac{1}{2} \frac{|k_l + k_r|}{|k_l - k_r|} - \frac{1}{2},$$ (13)

since the energy difference between the two sidebands corresponds to 1 phonon, and the sum corresponds to $2\bar{n}+1$ phonons[16,36,41,42]. We further verify that the power difference between the two sidebands is a constant by integrating Equation (11) (see Fig. 2(d, left)).

Alternatively, it is possible to integrate the measured and normalized heterodyne spectrum directly. We perform the integration by summing all the power spectrum near the mechanical peaks, with the noise floor $n_j$ subtracted. Let the summed results for the two sidebands be $s_l$ and $s_r$, the phonon number is then given by

$$\bar{n} = \frac{1}{2} \frac{|s_l + s_r|}{|s_l - s_r|} - \frac{1}{2}.$$ (14)

Due to the finite integration range and the broadening of the mechanical spectrum, the power difference between the two sidebands is no longer a constant. We compare it to the theoretical value, obtained by integrating the inferred mechanical spectrum from the homodyne measurement and within the corresponding frequency range. They show good agreement (cf. Fig. 2(d, right)).

## Feedback filter design

In our device, the targeted high-$Q_M$ mode is the fundamental mode. Other mechanical modes are far away, in particular, compared to other mechanical structures based on phononic crystals[13,47]. However, high-order out-of-plane modes still exhibit relatively large optomechanical coupling in our structure. In the presence of the signal transmission and processing delay, which is comparable to the oscillation period of the mechanical modes, an extra phase lag is therefore introduced. We tune the filter to have a phase response such that other mechanical modes are not heated, as described in detail in ref. 47. At different temperature, the mechanical frequencies shift slightly, and we tune the filter separately. The total feedback delay is tuned to 640 ns (with LHe cooling) and 680 ns (with LN$_2$ cooling).

## Data availability

Source data for the plots are available on Zenodo: https://doi.org/10.5281/zenodo.8172703.

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

## Acknowledgements

We would like to thank Ralf Riedinger and Sungkun Hong for their valu-able discussions. We also acknowledge assistance from the Kavli Nano-lab Delft. This work is supported by the European Research Council (ERC CoG Q-ECHOS, 101001005), and by the Netherlands Organization for Scientific Research (NWO/OCW), as part of the Frontiers of Nanoscience program, as well as through a Vrij Programma (680-92-18-04) grant. J.G. gratefully acknowledges support through a Casimir Ph.D. fellowship.

## Author contributions

J.G. and S.G. planned the experiment. J.G. performed the device design and sample fabrication. J.G., J.C., and X.Y. performed the measure-ments. J.G. and S.G. analyzed the data and wrote the manuscript with input from all authors. S.G. supervised the project.

## Competing interests

The authors declare no competing interests.
