## [Peer Review File · Nature Communications]

Active-feedback quantum control of an integrated low-frequency mechanical resonatorREVIEWER COMMENTS

Reviewer #1 (Remarks to the Author):

This manuscript by Guo et al describes experiments, in which a mechanical resonator mode of a fractal-like tree of silicon nitride strings (cf Ref [34]) is cooled to low occupations by active feedback (cf Ref [13]). The sensitive motion detection required for this purpose is implemented with an integrated photonic crystal cavity, whose near field is affected by the resonator's motion (cf Ref [46]).

The work might be seen as a combination of known elements, but I would still say the results are beautiful and noteworthy, not least from the point of view of a platform for future experiments (more on that below).

It is also a technically thorough study, in which two different measurements (in-loop and out-of-loop) are compared in order to give further confirmation of the results.

I therefore support publication in Nature Comms pretty much as is, with only a few minor comments and questions as listed here below.

1 - in the introduction the authors write that this is an easy-to-use experimental setup. This may be true after the fabrication is completed, which involves a manual step, if I am not mistaken. Also, for sensing, I presume a sample would be placed on the SiN resonator. How easy would this be? For example, how wide is the SiN string and can samples easily be placed there? Maybe the authors can qualify this statement of "easy use" further.

2 - The quality factors are good, but not as exorbitant as often reported recently. Is this a soft-clamped resonator? (If so, state in the manuscript). Are these levels of Q expected, then? The authors say that the design is inspired by Ref [34]. What did they do differently and why? Use the space in the supplement to fully describe the geometry of the resonator, potentially with a figure. The string width should also be stated in the main manuscript. Furthermore, with the small gap to the photonic crystal, could squeeze film damping play a role?

3 - The issue with the parasitic detection of the feedback actuation (laser intensity modulation) in the measurement channel is important enough to be mentioned in the main manuscript. The authors argue in the supplement that the effect is small, which they know from fitting the parameter epsilon. Give an explicit value of epsilon (in the main manuscript) as well as the loop delay (in the supplement).

4 - "The light merges with the cooling beam only after the intensity EOM, thus it is unaffected by our cooling modulation." It is not clear, even in the context, what this sentence means. Please rephrase / expand.

5 - Which laser sources are used in the experiments — please state. In case these are diode lasers: can it be excluded that laser phase noise leads to significant parasitic sideband asymmetry (see Ref [41])? Could be briefly discussed in the supplement.

Reviewer #2 (Remarks to the Author):

Reviewer report for "Active-feedback quantum control of an integrated, low-frequency mechanical resonator"

The manuscript entitled "Active-feedback quantum control of an integrated, low-frequency mechanical resonator" by Guo et al. demonstrates optical measurement-based feedback cooling of an integrated optomechanical system consisting of a low-frequency mechanical oscillator at 1.1 MHz, evanescently coupled to a photonic crystal cavity with a FWHM line width of 8.8 GHz. Starting from a cryogenically pre-cooled mode temperature of 18 Kelvin a minimum phonon occupancy of 0.76 is reached, signifying cooling of the mechanical resonator to its motional quantum ground state. Starting from a pre-cooled environment at 77 Kelvin a phonon occupancy of 3.5 is reached, approaching the mechanical quantum ground state. The authors claim that the reported system, operating deep in the unresolved sideband regime, and feedback cooling technique is well suited for sensing applications operating at the quantum limit.

The work presented in the manuscript builds on previously reported mechanical and optical devices and well-established measurement and analysis techniques in the field of cavity optomechanics. The manuscript is well structured and written, and the deployed experimental procedure is clearly described, except for a couple of misoriented beamsplitters in the schematic representations of the setup (two rightmost ones in Fig. 2a and rightmost one in Fig. 3a). The measurement data and analyses is clearly presented and adequately supports the conclusions in the manuscript.

Despite the unquestionable validity of the work and clarity of the presentation, a concern is whether the significance of the results is strong enough for publication in Nature Communications. Like the authors point out in the manuscript, they have themselves previously reported on the deployed integrated optomechanical device in Ref 46. Likewise, measurement-based feedback cooling to the motional quantum ground state has also previously been demonstrated for an optomechanical system operating in the sideband-resolved regime, cf. Ref. 13 by Rossi et al. In the introduction of the present manuscript the authors write: "Several seminal demonstrations of reducing the phonon occupancy below 1 have been achieved over the past years, often in the sideband-resolved limit where the cavity line width is smaller than the mechanical frequency [8-16]". However, in Ref. 13, the optical cavity linewidth and mechanical frequency were, respectively, 15.9 MHz and 1.14 MHz, placing it in the sideband-resolved regime, and here a phonon occupancy of 0.29 was achieved starting from a pre-cooled environment temperature of about 10 Kelvin. The device reported on by the authors clearly operate much deeper inside the sideband-resolved regime but that appears to be more of an engineering step.

While the device deployed by the authors does indeed represent a significant step forward towards an integrated sensor operating at the quantum level, a demonstration of such functionalisation of the device is missing. A description of a concrete scenario in which the device could feasibly be used as an "easy-to-use quantum technology" in a "real-world application", as the authors write in the second to last paragraph of the manuscript, supplemented by an analysis clearly showing that operation at the quantum level would offer a practical advantage, would substantiate the claimed application and the significance of the work.

Reviewer #3 (Remarks to the Author):

The manuscript of Guo et al. reports ground-state cooling of a nanostring mechanical resonator via measurement-based feedback.

Macroscopic mechanical objects were cooled to their ground states before only in a handful of experiments, and the manuscript reports the first instance of such cooling for an on-chip integrated device.

The one-dimensional mechanical resonators utilized in the work are high aspect-ratio strings made of stressed silicon nitride. Such structures can combine record-high mechanical quality factors and extremely low masses, and therefore are thought to be promising candidates for various sensing schemes. However, it had been a long-standing challenge to combine long nanostrings (which have the highest quality factors) with a high-efficiency optical readout that would allow harnessing their full potential, and, in particular, entering backaction dominated regime of position measurements. The authors demonstrated such a readout by integrating silicon nitride nanostrings with photonic crystals in a recent paper [Guo & Gröblacher Light Sci Appl 11, 282 (2022)], which enabled the present work.

I see the results reported by the authors as an important achievement and a landmark in a certain area. They go beyond the state of the art in terms of the capabilities of an integrated optomechanical sensor, and pave the way towards practical applications of high-stress nanostrings in classical and quantum sensing schemes.

Technically, the experiments reported in the manuscripts are performed with a high level of rigor (I do have a few questions though, see below), and the text is clearly written. The sideband asymmetry measurements are a particularly important piece of data that corroborates the author's main claims.

I recommend the manuscript to be published in Nature Communications after the small questions below are addressed and the missing details are given.

Particular remarks and questions:

1. A number of experimental details are missing in the text, which, I believe, should be mentioned.

What lasers were used in the experiments? At the operation frequencies as low as 1 MHz classical laser noises can be significant. Presumably, the authors verified that the laser noises contribute negligibly to the measured spectra? Did the authors implement length-balancing for the homodyne and heterodyne detection setups to cancel the phase noises of the lasers? (In the text, the homodyne detection referred to as “balanced”, but it is unclear whether it is balanced only in terms of the amplitudes on the two photodetectors or also the optical path lengths.)

Adding to the above, what is the signal splitting ratio between the homodyne and the heterodyne for the data in Fig 2?

2. It is a very interesting observation that the quantum performance of the device at elevated temperatures (77K) is limited by the extraneous imprecision noise due to the low-Q modes of the photonic crystal. It would be informative for the community if the authors presented and compared broadband spectra of Brownian motion taken at low and high temperatures, if they have such data available.

3. The authors mention using a phase modulation tone for calibrating the displacements spectra (the tone seems to be missing from the experimental plots, so I suppose that it is outside of the range). Their optical cavities, however, have quite broad linewidths, which can make such a scheme susceptible to errors due to the residual amplitude modulation from the EOM. Was it a problem in the experiment? Were any precautions taken to counteract this effect?

4. What is the loss budget for the optical measurement chain? Out of the overall 49% detection efficiency, 78% is contributed by the optical cavity itself. What are the losses on the coupling fiber, the circulator, the beamsplitter, and the photodetector, respectively? This needs to be mentioned together with other experimental details.

5. The heterodyne spectra taken at 77 K (Fig 4b) seem to have some spurious peaks, which are absent in the data taken at lower temperatures. Is this only an effect of the temperature

on the device, or something else was different between the two measurements?

6. Since the mechanical properties are of the key importance for sensing applications, did the authors perform a simulation of the mechanical quality factor? Does the measured value of the mechanical Q agree with the theory, and was there any degradation due to the integration with the optical cavity?

Reviewer #1 (Remarks to the Author):

This manuscript by Guo et al describes experiments, in which a mechanical resonator mode of a fractal-like tree of silicon nitride strings (cf Ref [34]) is cooled to low occupations by active feedback (cf Ref [13]). The sensitive motion detection required for this purpose is implemented with an integrated photonic crystal cavity, whose near field is affected by the resonator's motion (cf Ref [46]).

The work might be seen as a combination of known elements, but I would still say the results are beautiful and noteworthy, not least from the point of view of a platform for future experiments (more on that below).

It is also a technically thorough study, in which two different measurements (in-loop and out-of-loop) are compared in order to give further confirmation of the results.

I therefore support publication in Nature Comms pretty much as is, with only a few minor comments and questions as listed here below.

We would like to thank the referee for their positive assessment of our work and the following comments and suggestions to improve our manuscript.

1 - in the introduction the authors write that this is an easy-to-use experimental setup. This may be true after the fabrication is completed, which involves a manual step, if I am not mistaken. Also, for sensing, I presume a sample would be placed on the SiN resonator. How easy would this be? For example, how wide is the SiN string and can samples easily be placed there? Maybe the authors can qualify this statement of "easy use" further.

Currently, the device fabrication involves manual assembling. That being said, the assembling process is fully motorized, and the manual part only involves controlling the motor stages on the computer. Conceptually, the assembling process – putting a photonic crystal above a mechanical resonator – is similar to putting components on a PCB, which can be extended to allow for fully automated device assembly. We therefore believe that is indeed "easy to use" in a fully automated fashion.

Similarly, putting a sample on the SiN resonator can be done with the same pick-and-place method, and how easy it is depends on the size of the sample. As our present work is a proof-of-concept experiment, we did not explore the boundary of the width of the SiN string. However, realizing a SiN string with different widths, or even different geometry to accommodate different samples or applications, is relatively straight forward and certainly an interesting subject that we will investigate in the future.

2 - The quality factors are good, but not as exorbitant as often reported recently. Is this a soft-clamped resonator? (If so, state in the manuscript). Are these levels of Q expected, then? The authors say that the design is inspired by Ref [34]. What did they do differently and why? Use the space in the supplement to fully describe the geometry of the resonator, potentially with a figure. The string width should also be stated in the main manuscript. Furthermore, with the small gap to

the photonic crystal, could squeeze film damping play a role?

Yes, it is a soft-clamped resonator. We added a clarifying sentence to the main text “our integrated device consists of a ‘soft-clamped’ mechanical resonator inspired by a fractal structure”. The quality factor is in line with our expectation. There is a lot of room to improve the quality factor, such as using thinner SiN membrane and adapting some more recently developed designs. However, even with our relatively modest quality factor, it is already sufficient to get close to the ground state.

Our design deviates in several significant ways from the design in Ref [34]. We want to keep our resonance frequency to around 1 MHz to reduce the impact from low frequency noise in the setup and, as well as effects of gas damping. As shown in Figure 2 in Ref [34], the quality factor of the original design drops rapidly at higher frequency, and the advantage of the design over a simple beam diminishes above 600 kHz. This is why we developed our new design: in the original design, one tether is connected to two tethers. A large angle between the two child-tethers is required to achieve soft-clamping, which simultaneously however reduces the internal stress in the SiN film. We thus came up with our geometry, where one unit is connected to 3 sub-units. The additional sub-unit helps maintaining the stress. This design consideration is mentioned in our previous paper (ref [46]).

An even more detailed description, including the design method, can be found in our previous work (ref [46]). To highlight this, we have changed the related sentence in the main text to “they are assembled using a pick-and-place method. Details of the device structure and the assembly method are reported in [46].”

Thus far, we have not studied whether squeeze film effect is a limiting factor. We observe a small Q factor drop after assembling the full structure, as reported in [46]. However, although the gap is small, the width under the photonic crystal (0.8 μm) is not large. The aspect ratio of the gap (width / gap size) is not big, which is why we do not expect that the squeeze film effect to play a significant role.

3 - The issue with the parasitic detection of the feedback actuation (laser intensity modulation) in the measurement channel is important enough to be mentioned in the main manuscript. The authors argue in the supplement that the effect is small, which they know from fitting the parameter epsilon. Give an explicit value of epsilon (in the main manuscript) as well as the loop delay (in the supplement).

In the experiment, we took precaution to minimize the parasitic detection and made sure that we detect the Y quadrature, and that it is orthogonal to the laser intensity modulation. This piece of information is now mentioned in the supplement (Section “Data processing”). The fitted ϵ_{fb} is typically between 0.5×10^{-6} and 1.5×10^{-6} , and the range is a result of the electronic drift of the homodyne setpoint. As suggested by the referee, we now explicitly mention the explicit value of epsilon in the main manuscript (added at the beginning of page 4). The typical value is also added to the Supplementary Information (Section “Data processing”).

The feedback loop delay is 640 ns with LHe cooling and 680 ns with LN2 cooling. We have added this to the “Feedback filter design” section.

4 - “The light merges with the cooling beam only after the intensity EOM, thus it is unaffected by our cooling modulation.” It is not clear, even in the context, what this sentence means. Please rephrase / expand.

As noted in ref [41], classical amplitude noise can lead to significant sideband asymmetry. If the heterodyne probe laser is modulated by the intensity EOM, which introduces classical information about the mechanical resonator and the measured noise, it might generate spurious asymmetry. In our setup, this is avoided by combining the cooling beam and the heterodyne probing beam after the intensity EOM. We have expanded the sentence in the manuscript for clarity.

5 - Which laser sources are used in the experiments — please state. In case these are diode lasers: can it be excluded that laser phase noise leads to significant parasitic sideband asymmetry (see Ref [41])? Could be briefly discussed in the supplement.

We use external cavity diode lasers and characterized both their amplitude and phase noise. The classical phase noise around the mechanical frequency is about 400x of the shot noise, which results in only a minor contribution to the sideband asymmetry ($\lesssim 5\%$). We have added a section in the Supplementary Information discussing the noise.

To be more precise, as mentioned in ref [41], the impact of laser phase noise, normalized to noise quanta, is scaled by $\frac{\Omega_M}{\kappa} 10^{-4}$ (see equation (A35), and $\frac{4 \Delta_H}{\kappa} \lesssim 1$ in our case, where Δ_H is the heterodyne beam detuning). This scaling factor, due to the sideband-unresolved limit, make the impact of the phase noise negligible. Furthermore, as noted in ref [41], laser phase noise increases the power of both sidebands equality, leading to a more conservative estimation of the phonon number through sideband asymmetry.

Reviewer #2 (Remarks to the Author):

Reviewer report for "Active-feedback quantum control of an integrated, low-frequency mechanical resonator"

The manuscript entitled "Active-feedback quantum control of an integrated, low-frequency mechanical resonator" by Guo et al. demonstrates optical measurement-based feedback cooling of an integrated optomechanical system consisting of a low-frequency mechanical oscillator at 1.1 MHz, evanescently coupled to a photonic crystal cavity with a FWHM line width of 8.8 GHz. Starting from a cryogenically pre-cooled mode temperature of 18 Kelvin a minimum phonon occupancy of 0.76 is reached, signifying cooling of the mechanical resonator to its motional quantum ground state. Starting from a pre-cooled environment at 77 Kelvin a phonon occupancy of 3.5 is reached, approaching the mechanical quantum ground state. The authors claim that the reported system, operating deep in the unresolved sideband regime, and feedback cooling technique is well suited for sensing applications operating at the quantum limit.

The work presented in the manuscript builds on previously reported mechanical and optical devices

and well-established measurement and analysis techniques in the field of cavity optomechanics. The manuscript is well structured and written, and the deployed experimental procedure is clearly described, except for a couple of misoriented beamsplitters in the schematic representations of the setup (two rightmost ones in Fig. 2a and rightmost one in Fig. 3a). The measurement data and analyses is clearly presented and adequately supports the conclusions in the manuscript.

We thank the positive assessment of our work by the referee and the following comments and suggestions to the manuscript. We also thank referee for pointing out the three misoriented beamsplitters. They are now corrected in the Figure.

Despite the unquestionable validity of the work and clarity of the presentation, a concern is whether the significance of the results is strong enough for publication in Nature Communications. Like the authors point out in the manuscript, they have themselves previously reported on the deployed integrated optomechanical device in Ref 46. Likewise, measurement-based feedback cooling to the motional quantum ground state has also previously been demonstrated for an optomechanical system operating in the sideband-resolved regime, cf. Ref. 13 by Rossi et al. In the introduction of the present manuscript the authors write: "Several seminal demonstrations of reducing the phonon occupancy below 1 have been achieved over the past years, often in the sideband-resolved limit where the cavity line width is smaller than the mechanical frequency [8-16]". However, in Ref. 13, the optical cavity linewidth and mechanical frequency were, respectively, 15.9 MHz and 1.14 MHz, placing it in the sideband-resolved regime, and here a phonon occupancy of 0.29 was achieved starting from a pre-cooled environment temperature of about 10 Kelvin. The device reported on by the authors clearly operate much deeper inside the sideband-resolved regime but that appears to be more of an engineering step.

As mentioned by the referee, this work is based on devices we developed previously and reported in ref. 46, and our system is much deeper in the sideband-resolved regime comparing to ref. 13. Here, we would like to mention some advances to ref. 46 and some more differences comparing to ref. 13.

In ref. 46, we demonstrated the feasibility and the method of fabricating the integrated optomechanical device. Issues were also shown in ref. 46, such as the increased noise floor due to the motion of the photonic crystal. In this work, we experimentally demonstrate that our devices can actually be used in practical applications (creating delicate state where the mechanical motion is below 1 phonon).

Comparing to ref. 13, we are much deeper in the sideband-unresolved regime. This has practical benefits. For example, at low frequency, such as 1 MHz, typical lasers can have high phase noise, which might adversely affect some measurements. As pointed out by referee #1 and #3 and ref. 41, sideband-asymmetry measurement is such an example. However, operating deeply in the sideband-unresolved regime, the laser phase noise can be mitigated as the phase quadrature of the input light only weakly couples to the mechanical resonator. The coupling vanishes at the limit where the detuning is 0. This allow us to measure sideband-asymmetry, an important hallmark where our device is operated in the quantum level, in the presence of significant classical laser phase noise. We note that the sideband asymmetry is not shown in ref 13. Furthermore, a fully integrated optomechanical device in our work would be much easier to use in practice.

While the device deployed by the authors does indeed represent a significant step forward towards an integrated sensor operating at the quantum level, a demonstration of such functionalisation of the device is missing. A description of a concrete scenario in which the device could feasibly be used as an "easy-to-use quantum technology" in a "real-world application", as the authors write in the

second to last paragraph of the manuscript, supplemented by an analysis clearly showing that operation at the quantum level would offer a practical advantage, would substantiate the claimed application and the significance of the work.

In our work, we have demonstrated ground-state cooling and sideband asymmetry of an optomechanical system, which are the very first steps towards real-world quantum applications. As there are several real-world applications that could be potentially benefited from these significant advances, a detailed analysis of all of them lies beyond the scope of our current work. We have however listed a few in the second to last paragraph, and we have added several more in the revision (ref 7, 22, 25, 26, 58, 60, 61, 63, 64, 66). However, we do agree with the referee that discussing a concrete example would help in putting our work into context. In particular, with the large quantum cooperativity and the ability to have low thermal occupation, our system has great potential in building improved quantum transducers with no added classical noise, even at temperatures where the mechanical mode would typically have large thermal occupation. This could relax the requirements for building such transducers significantly and could find direct applications in the optical readout and control of qubits, for example. As suggested by the referee, we have highlighted this specific application in the revised manuscript.

Reviewer #3 (Remarks to the Author):

The manuscript of Guo et al. reports ground-state cooling of a nanostring mechanical resonator via measurement-based feedback.

Macroscopic mechanical objects were cooled to their ground states before only in a handful of experiments, and the manuscript reports the first instance of such cooling for an on-chip integrated device.

The one-dimensional mechanical resonators utilized in the work are high aspect-ratio strings made of stressed silicon nitride. Such structures can combine record-high mechanical quality factors and extremely low masses, and therefore are thought to be promising candidates for various sensing schemes. However, it had been a long-standing challenge to combine long nanostrings (which have the highest quality factors) with a high-efficiency optical readout that would allow harnessing their full potential, and, in particular, entering backaction dominated regime of position measurements. The authors demonstrated such a readout by integrating silicon nitride nanostrings with photonic crystals in a recent paper [Guo & Gröblacher Light Sci Appl 11, 282 (2022)], which enabled the present work.

I see the results reported by the authors as an important achievement and a landmark in a certain area. They go beyond the state of the art in terms of the capabilities of an integrated optomechanical sensor, and pave the way towards practical applications of high-stress nanostrings in classical and quantum sensing schemes.

Technically, the experiments reported in the manuscripts are performed with a high level of rigor (I do have a few questions though, see below), and the text is clearly written. The sideband asymmetry measurements are a particularly important piece of data that corroborates the author's main claims.

I recommend the manuscript to be published in Nature Communications after the small questions below are addressed and the missing details are given.

We thank the referee for the very positive assessment and the following comments and suggestions to improve our manuscript.

Particular remarks and questions:

1. A number of experimental details are missing in the text, which, I believe, should be mentioned.

What lasers were used in the experiments? At the operation frequencies as low as 1 MHz classical laser noises can be significant. Presumably, the authors verified that the laser noises contribute negligibly to the measured spectra? Did the authors implement length-balancing for the homodyne and heterodyne detection setups to cancel the phase noises of the lasers? (In the text, the homodyne detection referred to as "balanced", but it is unclear whether it is balanced only in terms of the amplitudes on the two photodetectors or also the optical path lengths.)

Adding to the above, what is the signal splitting ratio between the homodyne and the heterodyne for the data in Fig 2?

We use a Santec TSL-510 for cooling and a New Focus TLB-6728 for the heterodyne detection. We added this information under "Experimental setup" in the Methods section.

As discussed in the reply to referee #1, we verified that the laser noise has no significant impact on the measured spectra. We add the section "Laser noise for sideband-asymmetry detection" in the Supplementary Information for a more complete description. We balanced the path length for both homodyne and heterodyne detection. This information is also added to Section "Experimental parameters and setup". We did not see excess (classical) noise when detuning the laser away from the cavity resonance in the spectrum.

As listed in Table I in the Supplementary Information, the cooling (homodyne) beam is 0.79 μW , and the heterodyne beam is 1.3 μW . The homodyne and heterodyne beam are not split from a common source. They are from different laser with different wavelengths. The two beams are separated by a narrow-band filter cavity, allowing for nearly perfect separation.

2. It is a very interesting observation that the quantum performance of the device at elevated temperatures (77K) is limited by the extraneous imprecision noise due to the low-Q modes of the photonic crystal. It would be informative for the community if the authors presented and compared broadband spectra of Brownian motion taken at low and high temperatures, if they have such data available.

The frequency of the low-Q photonic crystal mode (at around 8.5 MHz) is too far away and we did not make systematic measurement of such a broad frequency range. We see strong $1/f$ scaling of the noise floor in the measurement, in accordance with what we would expect based on room temperature data (see [46]).

3. The authors mention using a phase modulation tone for calibrating the displacements spectra (the tone seems to be missing from the experimental plots, so I suppose that it is outside of the range).

Their optical cavities, however, have quite broad linewidths, which can make such a scheme susceptible to errors due to the residual amplitude modulation from the EOM. Was it a problem in the experiment? Were any precautions taken to counteract this effect?

Due to the broad linewidth of the cavity, we use a phase calibration tone with a much higher frequency than the mechanical resonator (60 MHz). It is still much smaller than the cavity linewidth, thus the frequency difference with the mechanical resonator does not generate significant error. We still took several precautionary measures:

1. The response of the electronics, including the photodetector and data acquisition instruments, is not flat within this spectrum range. We use optical shot noise to calibrate the photodetector response.
2. We verify that the residual amplitude modulation is not a concern by detuning the laser away from the cavity. The power at the modulation frequency is typically at least 30 dB lower when the laser is detuned away.

We added more information in the Methods section (“Experimental setup”).

4. What is the loss budget for the optical measurement chain? Out of the overall 49% detection efficiency, 78% is contributed by the optical cavity itself. What are the losses on the coupling fiber, the circulator, the beamsplitter, and the photodetector, respectively? This needs to be mentioned together with other experimental details.

We added more details to Table I in the Supplementary Information. The detection efficiency we listed in the table does not include the cavity itself, and now we explicitly mention this in the table.

5. The heterodyne spectra taken at 77 K (Fig 4b) seem to have some spurious peaks, which are absent in the data taken at lower temperatures. Is this only an effect of the temperature on the device, or something else was different between the two measurements?

The mechanical resonator still has many higher order modes, many of which also have high optomechanical coupling. At 77 K, their thermal motions are still large, and they still sample a relatively large portion of the cavity bandwidth. Note also that the detection response of the cavity is not perfectly linear. The nonlinearity generates spurious peaks at the mixing frequencies. By contrast, when the device is further cooled down, the motion of the other modes are also reduced, thus the peaks become non-detectable. This nonlinearity due to the mechanical motion and the detection, though in a different parameter regime, has been studied in [R. Leijssen et al, Nat. Commun. 8, 16024 (2017)]. We added a clarifying sentence in the caption of the Figure.

Otherwise, between the two measurements the setup is the same. Only the transfer function of the feedback controller was tweaked for different temperatures to accommodate slightly different mechanical frequencies.

6. Since the mechanical properties are of the key importance for sensing applications, did the authors perform a simulation of the mechanical quality factor? Does the measured value of the mechanical Q agree with the theory, and was there any degradation due to the integration with the optical cavity?

The design of the mechanical resonator is guided by simulation, where internal (bulk) material damping is considered. The measured value is about half of the simulated mechanical quality factor. We note that there are three important factors that are not included in our simulation, which could

potentially explain the lower quality factor.

1. Radiation loss, where the energy leaks from the clamping to the substrate.
2. Surface loss, which becomes more important for thin materials (e.g., see [L. G. Villanueva and S. Schmid, Phys. Rev. Lett. 113, 227201 (2014)]).
3. The effect of the integration with the optical cavity. The integration with the optical cavity degrades the mechanical Q factor, and is reported in our previous paper (ref [46]). By allowing a larger gap between the mechanical resonator and the photonic crystal, it is possible to mitigate this issue.

Other changes:

1. Change the convention of ϵ_{pb} (affecting equation S6 - S8), such that it is independent of g_{pb} .
2. Mention the linewidth of the filter cavity we use.
3. Fix a typo ("metrology" in the 1st paragraph).
4. We split the text into Introduction, Results, Discussion and Methods section. We add titles and subtitles to each section.
5. We have moved "Data processing" and "Feedback filter design" from the Supplementary Information to the Methods section. We further added "Experimental setup" in the Methods section, as well as a title to Figure 1, and specified the meaning of the error bars in each figure.

REVIEWERS' COMMENTS

Reviewer #2 (Remarks to the Author):

I would like to thank the authors for addressing the points raised in the review reports in a very clear and structured manner and for clearly stating and highlighting the changes made to the manuscript. This made it straight forward to trace the revisions.

It was a pleasure re-reading the manuscript. The added clarifications and details on the experimental setup and methods as well as the elaboration on practical applications of the results provides for a very strong paper which I am pleased to recommend for publication in Nature Communications.

Upon re-reading, I stumbled over one minor little typo in the added section title on page 4, second column at the bottom. There is a "d" missing in "Grou*d*state colling with improved detection efficiency".